# Synergistic Effect of Diet and Physical Activity on a NAFLD Cohort: Metabolomics Profile and Clinical Variable Evaluation

**DOI:** 10.3390/nu15112457

**Published:** 2023-05-25

**Authors:** Francesco Maria Calabrese, Giuseppe Celano, Caterina Bonfiglio, Angelo Campanella, Isabella Franco, Alessandro Annunziato, Gianluigi Giannelli, Alberto Ruben Osella, Maria De Angelis

**Affiliations:** 1Department of Soil, Plant and Food Science, University of Bari Aldo Moro, 70126 Bari, Italy; francesco.calabrese@uniba.it (F.M.C.); alessandro.annunziato@uniba.it (A.A.); maria.deangelis@uniba.it (M.D.A.); 2National Institute of Gastroenterology S. De Bellis, IRCCS Research Hospital, Via Turi 27, 70013 Castellana Grotte, Italy; catia.bonfiglio@irccsdebellis.it (C.B.); angelo.campanella@irccsdebellis.it (A.C.); isabella.franco@irccsdebellis.it (I.F.); gianluigi.giannelli@irccsdebellis.it (G.G.)

**Keywords:** Mediterranean diet, VOCs, NAFLD, liver disease, clinical impacting variables, rotating factor analysis

## Abstract

Together with its comorbidities, nonalcoholic fatty liver disease (NAFLD) is likely to rise further with the obesity epidemic. However, the literature’s evidence shows how its progression can be reduced by the administration of calorie-restrictive dietary interventions and physical activity regimens. The liver function and the gut microbiota have been demonstrated to be closely related. With the aim of ascertaining the impact of a treatment based on the combination of diet and physical activity (versus physical activity alone), we recruited 46 NAFLD patients who were divided into two groups. As a result, we traced the connection between volatile organic compounds (VOCs) from fecal metabolomics and a set of statistically filtered clinical variables. Additionally, we identified the relative abundances of gut microbiota taxa obtained from 16S rRNA gene sequencing. Statistically significant correlations emerged between VOCs and clinical parameters, as well as between VOCs and gut microbiota taxa. In comparison with a physical activity regimen alone, we disclose how ethyl valerate and pentanoic acid butyl ester, methyl valerate, and 5-hepten-2-one, 6-methyl changed because of the positive synergistic effect exerted by the combination of the Mediterranean diet and physical activity regimens. Moreover, 5-hepten-2-one, 6-methyl positively correlated with *Sanguinobacteroides*, as well as the two genera *Oscillospiraceae-UCG002* and *Ruminococcaceae UCG010* genera.

## 1. Introduction

Closely linked to a group of liver-related disorders, NAFLD (nonalcoholic fatty liver disease) is a chronic liver disease whose severity spectrum leads to liver damage, a condition that worsens in nonalcoholic steatohepatitis (NASH) and, in its advanced symptomatic picture, it irreversibly evolves into cirrhosis or hepatocellular carcinoma (HCC) [1].

When NAFLD and obesity pathologies are compared, it is clear that these two diseases share important pathological traits, including insulin resistance, dysbiosis, dyslipidemia, and hypertension [2]. In a ‘chicken and egg’ causality scenario featuring NAFLD patients, the associated comorbidities principally are caused by or imply a metabolic imbalance, and their improvement, in turn, improves NAFLD symptomatology. Patients with biopsy-proven NAFLD exhibit an increased incidence of some major adverse cardiovascular events that principally include ischemic heart disease, stroke, congestive heart failure, and cardiovascular mortality [3]. Regardless of the histological disease severity and focusing on disease risk stratification, a recent paper identified different NAFLD subgroups and found that one had signatures aligning with cardiovascular diseases (CVDs) and genetic risk factors [4].

Moreover, liver function and gut microbiota have been demonstrated to be closely associated [5]. Importantly, in light of the way the gut and liver influence each other through the gut–liver axis, the gut microbiota has been reassessed in terms of its influence in both metabolic-associated fatty liver disease (MAFLD) as well as in NAFLD [6], and the gut–microbiota–brain axis was proven to change relative to the liver disease spectrum [7].

In an advanced disease status, liver inflammation becomes persistent and extends to blood vessels [8], and proinflammatory cytokines activate downstream multi-protein complexes that finely tune a mechanism that establishes the inflammation bases promoting disease progression [9].

The liver is actively involved in the inflammatory/sub-inflammatory-related processes and the maintenance of immune homeostasis [10].

In addition to an altered immune response marked by neutrophils [11], monocytes, and dendritic cells [12], NAFLD clinical presentation features hepatic insulin resistance that, in turn, is dependent on an impaired glycogen synthase enzyme activity [13].

Higher total bilirubin concentrations have been associated with higher liver stiffness values in alcohol-related liver disease patients [14] and have been found to be altered in NAFLD patients [15]. In nonalcoholic steatohepatitis, the fecal metabolome helps delineate both the functional readout of the taxa in the gut and the final product of host metabolic activities, mirroring host–microbiome interactions [16].

Host-derived factors, including bactericidal fluids produced by gastric glands and the liver, impact the gut microbiota in terms of microbial load and composition [16], and the effectiveness of a dietary approach based on the Mediterranean diet pyramid is crucial in NAFLD patient routine care [17].

In a previous paper, we found how a weighted dietary regimen based on a low glycemic index Mediterranean diet contributed to reducing NAFLD dysbiosis and increasing microbial community resilience. As supported by measuring the controlled attenuation parameter (CAP) marker of the steatosis levels, adopting an aerobic exercise program further ameliorated the dysbiosis status, as sustained by specific asset modification of health-promoting microbes [18].

To investigate the impact of the synergistic effect of combining physical activity and the Mediterranean diet versus physical activity alone, we used fecal metabolomics profiles as the basic framework to search for the association between volatile organic compounds (VOCs) and a subset of statistically selected biochemical/clinical variables.

Moreover, we traced the correlation between significant microbiota taxa harbored by the two analyzed groups and the panel of statistically significant VOCs.

## 2. Materials and Methods

### 2.1. Study Groups

In this study, subjects with NAFLD in both hospital and general practitioners’ settings were enrolled. The studied cohort included two patient subgroups belonging to the small towns of Castellana Grotte and Putignano (district of Bari, Apulia, Italy), recruited by the Laboratory of Epidemiology and Biostatistics of the National Institute of Digestive Diseases, IRCCS ‘S. de Bellis’, Castellana Grotte, Italy, during the period covering March 2015 to December 2016 [19]. NAFLD staging from moderate to severe was determined based on the controlled attenuation parameter (CAP) score, which is useful in quantifying hepatic steatosis. Specifically, the adherence to the CAP score allows for ranking and classifying patients into four pathologic categories: absent (<248), mild status (248–267), moderate (268–279), and severe status (≥280). The inclusion criteria we used were based on (i) a body mass index (BMI) greater/equal to 25 kg /m^2^; (ii) age over 30 years old < 60; and (iii) NAFLD degree from moderate to severe. The exclusion criteria included (1) overt cardiovascular disease and revascularization procedures; (2) stroke; (3) clinical peripheral artery disease; (4) current treatment with insulin or oral hypoglycemic drugs; (5) fasting glucose > 126 mg/dL, or casual glucose > 200 mg/dL; (6) alcohol intake greater than 20 g/day; (7) medical conditions that could affect participation in a nutritional intervention study; and (8) people following a special diet, involved in a weight loss program, who had experienced recent weight loss, or were unable to follow a diet for religious or other reasons. Details about lifestyle interventions, enrolment, and sample size estimation have been previously described [20]. In total, we used 46 fecal samples from patients that were randomized into two study categories, as follows: (1) physical program consisting of aerobic activity and resistance training or anaerobic activity (ATFIS n = 20), or (2) ATFIS combined with low glycemic index Mediterranean diet (LGIMD-ATFIS35, n = 26). This study was conducted in accordance with the Helsinki Declaration and approved by the Ethical Committee (Prot. n. 10/CE/De Bellis, 3 February 2015). This study was based on a randomized clinical trial recently used for a recently published parallel work [18,21], namely, the NUTRIATT project, and registered on ClinicalTrials.gov (Identifier: NCT02347696). Stool samples were retrieved from the above-mentioned and previously analyzed cohort and referred to both 45 and 90 days of treatment. As we were interested in disclosing group-specific statistically significant differences in terms of VOCs emerging from the comparison between patients adhering to a Mediterranean dietary regimen alone or combined with physical activity (both aerobic and anaerobic), the time variable was not considered for statistical stratification of samples.

All participants signed an informed consent before being enrolled in the study.

### 2.2. Dietary Intervention and Data Collection

No specific indications were given regarding the total calories to be consumed by patients that were on LGIMD but included foods marked by a low glycemic index, and saturated fats overall represented no more than 10% of total daily calories. More precisely, the LGIMD diet contained high levels of monounsaturated fatty acids (MUFA) from olive oil and omega-3 polyunsaturated fatty acids. A brochure format enriched with graphic explanations illustrated the diet guidelines to be followed.

To collect diary details, sociodemographic aspects, lifestyle, and medical history, patients were interviewed during the trial by nutritionists.

Physical activity regimen details were collected thanks to the validated International Physical Activity Questionnaire—Long Form (IPAQ-LF); whereas, to probe alcohol intake and eating behavior, the European Prospective Investigation into Cancer and Nutrition Food Frequency Questionnaire (EPIC FFQ) was used.

### 2.3. Fecal Metabolome Analysis

A total of 1 g of fecal sample was placed into 20 mL glass vials and sealed with a magnetic screw cap provided with PTFE/silicone septa. Ten μL of 4-methyl-2-pentanol was added into SPME vials (final concentration of 33 mg/L) as internal standard. After an equilibration for 10 min at 60 °C, SPME fiber (divinylbenzene/Carboxen/polydimethylsiloxane) was exposed to each sample for 40 min. The volatile organic compounds (VOCs) desorption was carried out for 3 min in a gas chromatography (GC) injector equipped with a quartz liner operating in splitless mode at 220 °C. A single-quadrupole mass spectrometer Clarus 680 (Perkin Elmer, Seer Green, Beaconsfield, UK) gas chromatograph equipped with an Rtx-WAX column (30 m × 0.25 mm i.d., 0.25 μm film thickness) (Restek Corporation, Bellefonte, PA, USA) and coupled to a mass spectrometer (MS) Clarus SQ8MS (Perkin Elmer) was used to identify the extracted compounds. The column temperature was set initially at 35 °C for 8 min, then increased to 60 °C at 4 °C min^−1^, to 160 °C at 6 °C min^−1^, and finally to 200 °C at 20 °C min^−1^ and held for 15 min. Electron ionization masses were recorded at 70 eV in the mass-to-charge ratio interval, *m*/*z* 34 to 350. The GC-MS system generated a chromatogram with peaks representing individual compounds. Each chromatogram was analyzed for peak identification using the National Institute of Standard and Technology 2008 (NIST) library. A peak area threshold of >1,000,000 and 85% or greater probability of match was used for VOC identification, followed by manual visual inspection of the fragment patterns when required. The relative concentration (µg/g of 4-methyl-2-pentanol) was esteemed by interpolating the relative areas versus the internal standard area.

### 2.4. Differential Abundance Analysis, Principal Component Analysis, and Orthogonal Projections for Latent Structure Discriminant Analysis and Clustering Analysis

We performed an exploratory factor analysis using the principal-component factor model. This allowed us to reduce the number of variables by describing linear combinations of the variables that contain most of the information and that admit meaningful interpretations.

To restrict the entire panel to factors with a strong clinical interpretation, we a priori picked eigenvalues equal to or greater than two. A graphical assessment of the eigenvalues was performed. We used post-estimation tools such as orthogonal rotation (varimax) to give more weight to large initial loadings. Finally, the Kaiser–Meyer–Olkin (KMO) measure of sampling adequacy was applied [22].

White’s nonparametric test corrected for multiple tests (Benjamini–Hochberg) was used for two group comparisons.

For VOC statistical analyses, we stratified samples based on the fed group by using partial least square discriminant analysis (PLSDA) and orthogonal projections to latent structures discriminant analysis.

Correlation analyses were based on corrected Pearson’s correlation test ‘cor’ run in the R environment “https://cran.r-project.org/web/packages/COR/index.html (accessed on 25 January 2023)”, and statistically significant correlations were graphically rendered with R’s ‘corrplot’ package.

### 2.5. Multivariable Association via the MaAslin2 R Package

Fresh fecal samples were collected from all enrolled subjects. According to the manufacturer’s instructions, the QIAamp FAST DNA Stool Mini Kit (Qiagen, Hilden, Germany) was used to extract the total bacterial stool DNA used for microbiota analyses. The final yield and quality of extracted DNA were checked by means of a NanoDrop ND-1000 spectrophotometer (Thermo Scientific, Waltham, MA, USA) and Qubit Fluorometer 1.0 (Invitrogen Co., Carlsbad, CA, USA). The 16S metagenomic analyses were performed at Genomix4life S.R.L. (Baronissi, Salerno, Italy). Specifically, 16S rRNA amplification, targeting the V3-V4 hypervariable region, was performed with the following primer couple—Forward: 5′-CCTACGGGNGGCWGCAG-3′ and Reverse: 5′-GACTACHVGGGTATCTAATCC-3′. Each PCR reaction was set based on the Metagenomic Sequencing Library Preparation (Illumina, San Diego, CA, USA). A negative control was included in the workflow, consisting of all the reagents used during the sample processing (16S amplification and library preparation) but not containing the sample. The resulting libraries were quantified using those as previously reported [18].

Raw fastQ files were inspected for base quality using FastQC software version 0.12.1 “https://www.bioinformatics.babraham.ac.uk/projects/fastqc/ (accessed on 25 January 2023)”. In silico bioinformatics analyses, including denoising, taxa assignment, and alpha and beta diversity, were computed with QIIME2 version 2020.8 (https://github.com/qiime2/qiime2 accessed on 25 January 2023) microbiome platform and nested plugins. More specifically, the q2-deblur plugin was used for the denoising step, and Shannon’s/Faith’s PD metrics were run based on the significance obtained using ad hoc computed statistics. The SILVA 138 SSU database (https://www.arb-silva.de/documentation/release-138/ accessed on 25 January 2023) was used to infer the taxonomy based on the taxa identification. All the final outputs not included in this paper are available upon request and will be provided in ‘.qzv’ file format. Raw fastQ files are part of the NCBI BioProject ID PRJNA816444.

## 3. Results

The adopted approach was useful in studying the two subpopulations of an Italian NAFLD patient cohort, i.e., ATFIS and LGIMD/ATFIS, which relied on a combination strategy that evaluated metabolomics volatile organic compounds and a set of detected biochemical clinical parameters. In the present study, physical activity and its combination with the Mediterranean diet were the two conditions we tested.

Stringent statistical measures were adopted to (i) obtain the most impactful biochemical clinical parameters that allowed for the recognition of simplified signatures related to the NAFLD pathology, (ii) detect the set of statistically significant VOCs leading to the stratification of samples and which differ between the two tested conditions, (iii) find statistically significant correlations between VOCs and clinical parameters, and (iv) find statistical correlations between VOC and 16S rRNA detected taxa from our previous paper.

### 3.1. Rotated Factor Loadings (Pattern Matrix) and Unique Variances of Biochemical Clinical Detected Variables

The complete matrix of biochemical detected parameters was reduced thanks to the pattern matrix analysis that reduced its complexity.

Table 1 reports the proportion of variable common variance that was not associated with the factors. We considered variables whose uniqueness was less than 0.5, where 1 represents commonality. Factor loadings represent both how the variables are weighted for each factor and explain the correlation between variables and factors. As reported in Appendix A, four out of the total 28 latent variables had an eigenvalue greater than two. Consequently, we decided to perform the subsequent factor analysis based only on this factor subset.

Based on this analysis, we were able to putatively collapse the latent variables into four factors that can be ascribed to different processes/profiles. Specifically, we can attribute the first factor to an inflammatory/sub-inflammatory process. The second is related to the metabolism of glucose and can be identified as ‘metabolic’. The third is related to hepatic pain, and we referred to it as ‘parenchymatous/hepatic’. Lastly, the fourth factor, which specifically accounts for total and direct bilirubin and hemoglobin, is involved in the prognosis of cardiovascular pathologies that are associated with NAFLD.

### 3.2. Detected Volatile Organic Compounds (VOCs)

A total of 127 VOCs were identified in our sample set composed of 26 ATFIS plus 20 LGIMD-ATFIS samples (Appendix A). The entire set of metabolites was inspected by orthogonal projections to latent structures discriminant analysis (OPLS-DA), and the plotted samples revealed the separation of the two group clouds (Figure 1A).

As evidenced by the OPLS-DA variable importance in the projection (VIP) plot (Figure 1B), the list of greatest-contributing VOCs includes cetene, 1-pentadecene, 1,5,9-Decatriene, germacrene B, caparratriene, methyl valerate, and 2-Heptanone 5-methyl. The cross-validation coefficients (Q2) produced by the permutation analysis between one predictive and orthogonal component supported the separation between the ATFIS and LGIMD-ATFIS groups (Figure 1C).

To better inspect those VOCs that allow for two-group separation, we ran a pairwise nonparametric test. More specifically, the volcano plot shown in Figure 2 reports statistically significant VOCs that emerged from a nonparametric Kruskal–Wallis test comparison versus a fold-change analysis. Noteworthy, 5-hepten-2-one_6-methyl-, cetene, perillene, 1-pentadecene, dimethyl_trisulfide, 159 decatriene_2358-tetr, anethole, 1-butanol_3-methyl-, and tridecane decreased in ATFIS group, whereas methyl valerate, butanoic acid methyl ester, 2-piperidinone, pentanoic acid ethyl ester, levomenthol, butanoic acid butyl ester, butanoic acid propyl ester, pentanoic acid butyl ester, and heptanoic acid ethyl ester were significantly decreased in the LGIMD-ATFIS group. Fold change and *p*-values for all the statistically significant metabolites are reported in Appendix A.

### 3.3. Correlation between VOCs and Clinical Parameters

To compare the fecal volatilome profile and clinical parameters of NAFLD patients, we used the total sets of significant VOCs and clinical parameters in a Pearson’s correlation analysis. Almost all the statically significant cross-correlations between the two groups of variables were positively significant correlations (*p*-value < 0.5). Caparratriene was positively correlated with insulin and C-peptide, while eosinophils were positively correlated with anethole and 1-pentadecene. Although marked with low correlation values, hemoglobin was significantly correlated with pentanoic acid butyl and ethyl ester, methyl valerate, and heptanoic acid ethyl ester. In turn, methyl valerate and butanoic acid methyl ester positively correlated with direct bilirubin (Figure 3).

### 3.4. Comparison between Statistically Significant VOCs and Taxa

The analyzed sample subset has already been characterized by our research group in a previous paper describing the taxonomic differences emerging from the comparison between LGIMD and a control diet based on CREA-AN (INRAN) diets while considering the impact of both aerobic and anaerobic physical activity regimens [18]. Here, our cohort subset of annotated taxa was inspected by the general linear model implemented in the ‘MaAsLin 2′ R package (microbiome multivariable association with linear models). As a result, the LGIMD and ATFIS/LGIMD group comparison returned nine different statistically significant genera (Appendix A). In detail, all the identified taxa, i.e., an uncultured genus from the Peptococcaceae family, Rikenellaceae RC9 gut group, Catenibacterium, *Oscillospiraceae-UCG002*, *Ruminococcaceae UCG010*, Family XIII AD3011 group, *Lachnospiraceae GCA-900066575*, *Haemophilus,* and *Sanguibacteroides* had a higher relative abundance in the group where the two activity programs and the Mediterranean diet intervention were combined. The complete list of significative taxa (q < 0.05) is reported in Appendix A.

To find a biological connection between metabolite profile and microbiota, statistically significant annotated taxa from our NAFLD patient cohort, selected from the MaAsLin regression model and VOC, were correlated. As a result, Pearson’s correlation returned several positive correlations, among which 159-Decatriene_2358-tetr and 1-pentadecene increased together with *Sanguibacteroides* and *Lachnospiraceae UCG002*.

Additionally, cetene positively correlated with this Lachnospiraceae genus. *Sanguibacteroides,* in turn, correlated with 5-hepten 2 one 6 methyl and perillene. These last two metabolites were significantly related to increased levels of *Lachnospiraceae UCG002* and *Ruminococcaceae UCG010*.

The highest positive correlation was found for the *Haemophylus* versus butanoic acid butyl ester. Pentanoic acid butyl ester and levomenthol were positively correlated with *Haemophylus* (Appendix A).

## 4. Discussion

The NAFLD patient biochemical signature is characterized by different risk factors, mainly accounting for the systemic dysregulation of lipid metabolism, inflammation, and diabetes [23]. Homeostasis recovery, along with the entire whole gut–liver axis and, consequently, symptom relief, have been proven to be directly connected with the gut microbiota [24]. Since we are interested in disclosing the synergistic effect of different lifestyle interventions, in previous work, we explored specific microbiota rearrangements in a NAFLD patient cohort subjected to different diet and physical activity regimens, both aerobic and anaerobic, and we confirmed these data by measuring the level of steatosis (CAP score) [18].

With the presented batch of data, we aimed to ascertain the impact of physical activity, whether it was aerobic or anaerobic, combined with a Mediterranean diet in patients affected by NAFLD. We correlated the resulting stool metabolomics profiles with biochemical clinical data. To accomplish this, we first made use of a statistical approach that allowed for the selection of specific biochemical parameters, and subsequently, we collapsed them into impacting factor classes. As a result, we identified variables related to (i) an inflammatory/sub-inflammatory process accounting for WBC, neutrophils, lymphocytes, monocytes, and eosinophils; (ii) a metabolic factor (glucose, HbA1c, C peptide, insulin); (iii) a parenchymatous/hepatic factor (hepatic pain as linked to GOT, GPT, GGT); and (iv) a factor in common with cardiovascular pathologies (total and direct bilirubin and hemoglobin). Although the temporal relationship between an elevated white blood cell count (WBC) and NAFLD is difficult to be determined, this inflammation marker was found to be significantly associated with incident NAFLD in different large-scale Asian longitudinal cohorts [25].

As in the case of other liver diseases, NAFLD onset and damage progression both involve innate and adaptive immunity, with a possible proinflammatory gene expression in liver Kupffer cells before evident hepatic inflammation is detected [26].

Among the variables composing the first factor, lymphocytes and myeloid cells (including monocytes, neutrophils, and eosinophils, among granulocytes) showed the highest factor loadings and uniqueness values. This statistical evidence supports the well-known phenotypic and functional changes in lymphocytes and peripheral leukocyte compartments, where neutrophils are the major inflammatory infiltrating cells. It is worth noting that eosinophils positively correlated with 1-pentadecene and anethole. This initially discussed correlation in evidence reveals how both these VOCs increased when the diet was coupled with physical activity.

A recent prospective study inspected the noninvasive volatile organic profile from the exhaled breath of asthma-affected patients and reported how a higher average concentration of 1-pentadecene was detected in neutrophilic versus eosinophilic asthma phenotypes, most likely in response to the anti-inflammatory treatment [27]. We could explain the correlation between eosinophils and pentadecane as a consequence of the activation of M2 macrophages; this process would alleviate liver injury by suppressing excessive immune responses [28]. Additionally, by inhibiting the activation of the TGF-β signal pathway, anethole exerts its protective effect on hepatic fibrosis and nonalcoholic steatohepatitis [29]. Additionally, this same study demonstrated a dose-dependent effect of anethole, capable of ameliorating liver injury in mice. Based on the impact of this aromatic compound on white adipose tissue browning, and its importance in the activation of brown adipose tissue, we can suppose a protective effect on LGIMD-ATFIS NAFLD patients too.

Two out of four variables included in the second factor, C-peptide and insulin, were related to metabolism. A second important positive correlation emerged between both these variables and caparratriene, a sesquiterpene, identified as the major bioactive compound present in ethanolic leaf extract and known for its inhibitory activity against lymphoid CEM leukemia cells [30]. In our analyses, the caparratriene compound emerged as the most scored OPLS-DA VIP in the LGIMD-ATFIS group. Higher fasting C-peptide and fasting insulin, emerging from our rotating factor analysis, were found in NAFLD patients, as reported by Han and colleagues [31], and insulin-resistant patients commonly exhibited increased insulin serum levels.

Variables belonging to the fourth factor, hemoglobin and direct bilirubin, positively correlated with fatty acid methyl- and ethyl esters (FAME and FAEE). FAMEs have been detected in plasma, and their presence has been correlated with liver dysfunction [32]. Specifically, pentanoic acid propyl ester (ethyl valerate), pentanoic acid butyl ester, methyl valerate, and heptanoic acid ethyl ester decreased when the Mediterranean diet and physical activity were combined and correlated with higher hemoglobin levels. As recently documented, NAFLD patients have high levels of serum circulating hemoglobin in comparison with healthy controls [33]. By acting as an antioxidant, the high hemoglobin levels seem crucial in counteracting NAFLD’s adverse effects [34].

It is noteworthy that NAFLD has been associated with cognitive impairment [35] and hyperammonemia [36]. The hyperammonemic status, which marks liver cirrhosis, is dependent on BCAA-related metabolite methyl valerate that is significantly correlated with both HOMA-IR and the fat-free mass index (FFMI) [37]. Methyl valerate, the metabolic product of isoleucine, has been proven to stimulate lipid peroxidation in insulin-resistant subjects at elevated concentrations [38]. Since the methyl valerate concentration increased in our ATFIS group, we can hypothesize a protective effect resulting from the synergistic combination of the Mediterranean diet and physical activities. At the same, the pentanoic acid ethyl ester and butyl ester, also known as ethyl and butyl valerate, follow the same trend. In line with our data, these fatty esters, together with other fecal ester VOCs, have been found to be more abundant in obese NAFLD cohort patients [39].

Moreover, butanoic acid methyl ester and methyl valerate positively correlated with direct bilirubin, one of the variables included in the fourth identified factor. As previously mentioned, ammonia-related disorders interfere with bilirubin metabolism in hepatocytes, and this would sustain the correlation with methyl valerate in a pathologic status featured by hyperammonemia.

Focusing on the correlation between significant taxa and VOCs, a positive statistical value marked the comparison of *Haemophilus* with both butanoic acid butyl ester and pentanoic acid butyl ester. This genus, which in our sample cohort was decreased in the LGIMD-ATFIS group, is one of the taxa that most significantly contributed to lipopolysaccharide production, and its increased abundance was found in hepatocellular carcinoma [40]. Noteworthy, fecal ester compounds are increased in NALFD patients [39], and this reflects the protective effect of the combined intervention we proposed that is aimed at restoring gut–liver axis eubiosis.

In line with other literature evidence [41], we found the genus *Sanguinobacteroides* and the two genera *Oscillospiraceae-UCG002* and *Ruminococcaceae UCG010* increased in LGIMD-ATFIS and positively correlated with various VOCs, including 5-hepten-2-one, 6-methyl. The reverse hydroxyl aldehyde condensation reaction of citral, triggered by proteins and amino acids, produces 5-hepten-2-one, 6-methyl (or methyl heptanone), and acetaldehyde [42]. More specifically, this reaction happened in the presence of polar amino acids. Since a significant disproportion of polar amino acid serum levels was found in subjects with NAFLD and liver fibrosis [43], we can speculate on the possibility that this condition would impact the metabolism of 5-hepten-2-one, and 6-methyl as well. Moreover, a recent study reported how lemongrass oil, whose major component is citral, reduces enzymatic activities and lowers liver oxidative stress [44], as well as reduces liver injury in high-fat diet-induced NAFLD mice [45]. Thus, we show a positively increased effect exerted by this molecule in LGIMD-ATFIS-treated patients. Prompted by a high positive correlation, the concomitant increase in the *Sanguibacteroides* genus in LGMID-ATFIS, found to be negatively correlated with AST, ALT, GGT, and uric acid levels [46], suggests a positive impact of the combined intervention on eubiosis status restoration.

Finally, we found how increased levels of the monoterpenoid perillene, a component of the essential oil with antibacterial and antitumor effects [47], and cetene [48] were both positively correlated with increased relative abundances of *Oscillospiraceae-UCG002* and *Ruminococcaceae UCG010*, both important in improving gastrointestinal barrier integrity.

In conclusion, the present study indicates how specific metabolites and taxa significantly differed in NAFLD patients under a dietary/physical activity combined intervention. From a biological perspective, the correlation between specific VOC and biochemical variables allowed us to investigate a panel of potential markers linked to host response mechanisms and metabolic processes.

## 5. Conclusions

To investigate whether the synergistic impact obtained by the union of diet and physical activity regimen was capable of restoring NAFLD patient cohort dysbiosis versus physical activity alone, we detected significant changes in taxa relative abundances and VOC concentrations, and we inspected downstream the relationship between them. Moreover, metabolomics profiles were connected to clinically detected variables, gathered in factor classes, thanks to a robust statistical approach accounting for rotating factor analysis. The elucidated metabolomics signature included ethyl- and methyl valerate, pentanoic acid butyl ester, and heptanoic acid ethyl ester that decreased in those patients who underwent the combined treatment, whereas cetene, perillene, 1-pentadecene, and anethole increased. Concurrently, higher relative abundances of *Sanguinobacteroides*, *Oscillospiraceae-UCG002,* and *Ruminococcaceae* genera differentiated the LGIMD-ATFIS study group.

Although the two sampling times were sufficient to delineate statistically significant differences among the groups, the overall time extension of the trial, limited to 90 days, may be considered a limiting factor of the present study. Moreover, through us being interested in the comparison between the two regimens, we analyzed the samples after the useful period required to register an effect both in terms of volatilome and microbiota-altered profiles. Thus, neither the length of the intervention (45 or 90 days) was considered, nor were samples at the baseline (T0) included in the cohort.

In future work, a metatranscriptomics approach on a multi-centric NAFLD patient cohort would help elucidate the major functions exerted by the key taxa that crucially impact inflammation and other linked comorbidity processes, thus reducing hepatic pain. The statistical elaboration of data resulting in the identified factors is, in fact, a model that can be useful for comparing other NAFLD cohorts from other countries.

## Figures and Tables

**Figure 1 nutrients-15-02457-f001:**
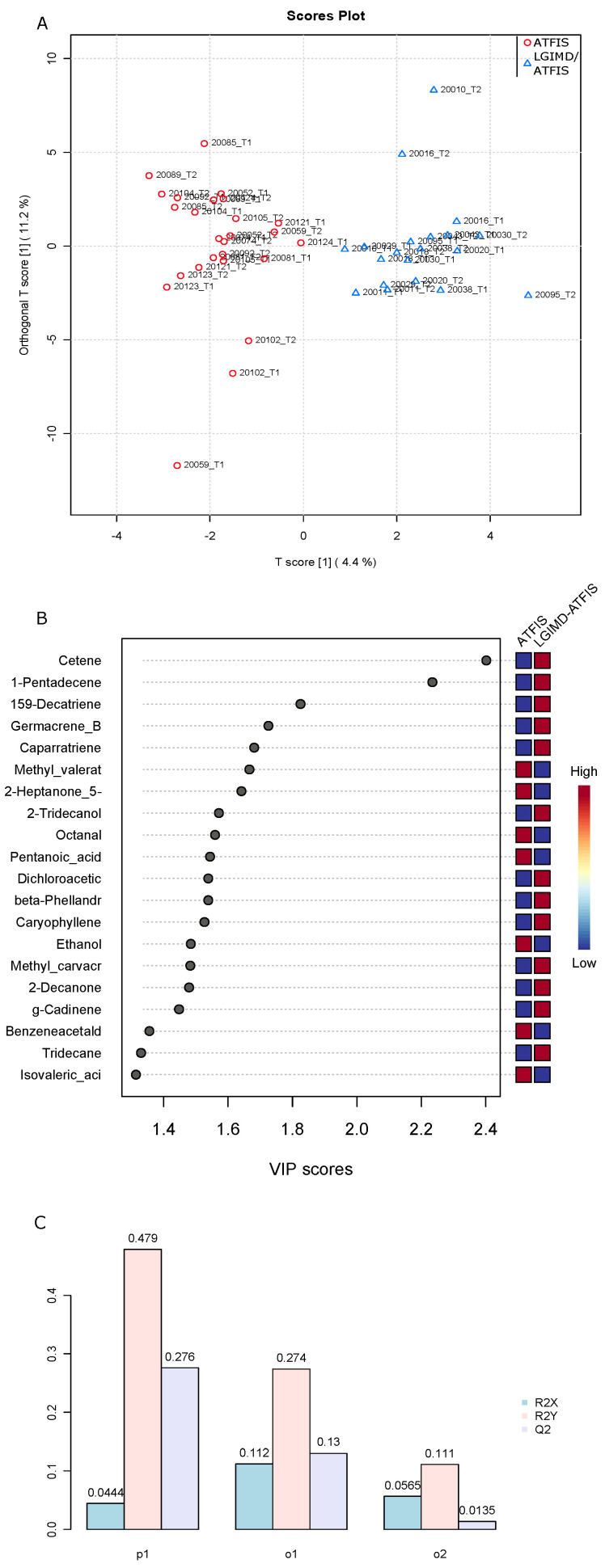
OPLS-DA analysis based on VOC abundance normalized matrix. (**A**) T score and orthogonal T score have been used to plot samples onto a bidimensional graph. Red and blue colors indicate ATFIS and LGIMD-ATFIS group, respectively. (**B**) OPLS-DA variable importance in projection (VIP) plot has been computed based on the complete panel of detected VOCs. Mostly important metabolite features identified by OPLS-DA are ranked on the top. Blue and red boxes on the right indicate relative concentration of corresponding VOC for samples belonging to AFTIS and LGIMD-ATFIS groups. (**C**) Cross validated Q2/R2X/R2Y coefficients produced as a result of a permutation analysis between one predictive (p1) and two orthogonal (o1, o2) components.

**Figure 2 nutrients-15-02457-f002:**
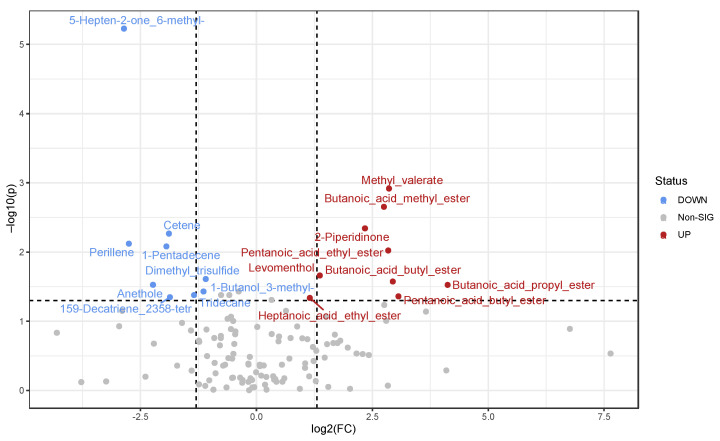
VOC volcano plot from fold change analysis. Statistically significant VOCs emerged from a nonparametric Wilcoxon rank-sum test combined with fold change (FC) analysis. Due to the chosen comparison direction ATFIS/LGIMD-ATFIS increased and decreased metabolite concentrations in ATFIS groups have been marked as down (blue) and up (red), respectively. The −log10 (*p* values) is meaningful of the level of significance of each VOC and has been plotted versus the log2 fold change. It represents the difference between the levels of expression for each VOC between the two groups.

**Figure 3 nutrients-15-02457-f003:**
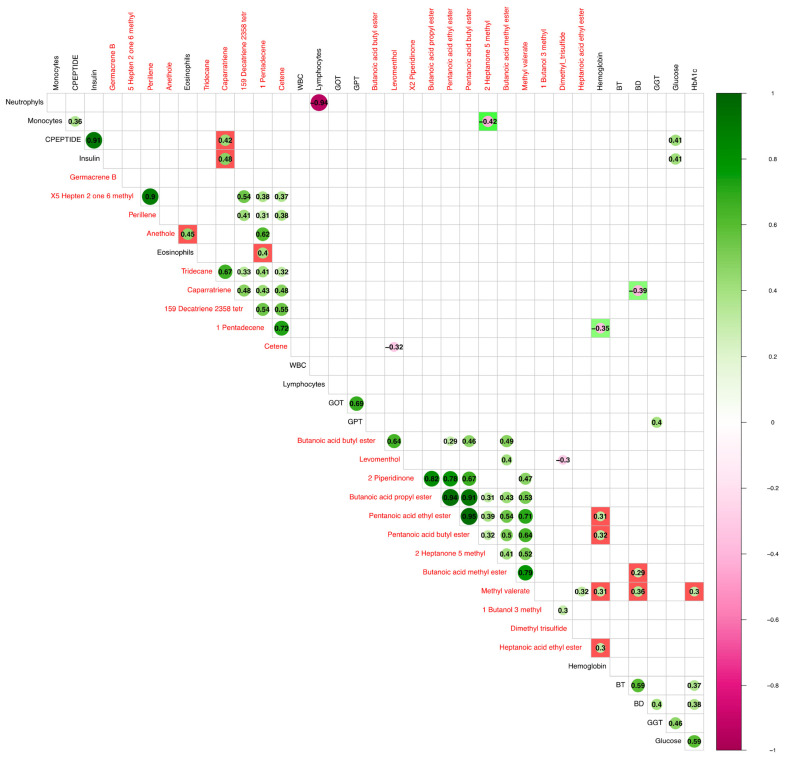
Pearson correlation analysis between statistically significant VOCs and biochemical clinical detected parameters. Biochemical clinical parameters and volatile organic compounds that significantly differed in the ATFIS and LGIMD/ATFIS groups were correlated by meaning of a Pearson’s test and only statically significant comparisons (*p* < 0.05) were plotted. Black and red fonts mark biochemical parameters and VOCs, respectively. In the legend scale bar, green and purple colors are meaningful of positive and negative correlation, respectively. Cross group paired variables were highlighted with red and green background colors, indicating positive and negative significative cross correlations, respectively.

**Table 1 nutrients-15-02457-t001:** Variable common variance associated with factors. Four factors out of 28 latent variables were chosen based on eigenvalue (>2). Based on their uniqueness, factor loadings evidenced the most impacting variables (highlighted via the bold font).

Variable	Factor1	Factor2	Factor3	Factor4	Uniqueness
RBC	0.4276	0.0287	0.1607	0.2875	0.7078
hemoglobin	0.2019	−0.0004	0.2233	**0.5525**	**0.6041**
Platelets	0.2915	−0.0773	0.2949	−0.6362	0.4173
WBC	**0.8994**	0.0361	−0.0157	−0.1788	**0.1575**
neutrophils	**0.7489**	0.0059	−0.1133	−0.0779	**0.4202**
lymphocytes	**0.5892**	0.1021	0.1975	−0.3466	**0.4832**
monocytes	**0.6879**	0.1248	−0.0648	0.1637	**0.4803**
eosinophils	**0.6177**	−0.205	0.1259	−0.1964	**0.522**
basophils	0.5387	−0.026	0.1105	0.1217	0.6821
glucose	−0.0541	**0.7698**	−0.1217	0.0499	**0.3871**
HbA1c	−0.2133	**0.7414**	0.0225	−0.1435	**0.3837**
UREA	−0.0032	0.1894	−0.2616	0.2071	0.8528
Creatinine	0.0352	0.1035	0.0187	0.5968	0.6315
eGFR	−0.1503	−0.1806	−0.1691	−0.1227	0.9012
BT	−0.0903	−0.0983	0.2386	**0.6759**	**0.4684**
BD	−0.2108	0.0297	0.1298	**0.7783**	**0.3321**
GOT	−0.0714	0.0742	**0.8491**	0.013	**0.2682**
GPT	0.0603	0.0501	**0.8683**	0.0991	**0.23**
GGT	−0.0316	0.0312	**0.6658**	0.0636	**0.5506**
Iron	−0.0772	−0.1194	0.1474	−0.103	0.9474
Cholesterol	0.1812	−0.3587	0.1512	−0.3082	0.7206
HDL	−0.3543	−0.4732	−0.2486	−0.0437	0.5868
FT3	0.1707	−0.0438	0.4859	0.0339	0.7317
FT4	0.0086	0.0938	−0.1718	0.0823	0.9549
Cortisol	−0.2473	−0.2376	-0.1526	0.1537	0.8355
C-PEPTIDE	0.177	**0.7786**	0.0823	0.107	**0.3443**
Insulin	0.1228	**0.7452**	0.1835	0.0024	**0.396**

## Data Availability

The obtained 16S rRNA fastQ sequences are available from the NCBI BioProject database. The project submitted entry refers to Submission ID: SUB11191038; BioProject ID: PRJNA816444. All data and results are available within the provided material.

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
