# Peer review of "Synergistic Effect of Diet and Physical Activity on a NAFLD Cohort: Metabolomics Profile and Clinical Variable Evaluation"

_nutrients, 2023, doi:10.3390/nu15112457_

Round 1
Reviewer 1 Report
ABSTRACT
It is surprising that the two main words of the title of the manuscript (“diet” and “physical activity”) are not in the aim of the abstract. As a reviewer, it is expected a correspondence between the title of the manuscript, the aim and the conclusion.
In the abstract, there is no mention to the number of subjects involved in the analysis or their characteristics except saying “an Italian NAFLD patient cohort composed of patients following a physical activity intervention alone or coupled with the Mediterranean diet”.
The acronym VOCs appears in the abstract without definition. VOCs it is not defined until the line 115 (Volatile Organic compounds). The term VOCs was used six times in the text before providing a definition of the acronym.
The conclusion of the abstract is very vague, with no connection to the title of the manuscript. If the title of the manuscript is on the synergistic effect of diet and physical activity, it is expected to see in the conclusion whether this putative interaction is synergistic or not, and whether there are VOCs involved in such putative synergistic effects. According to this referee, it is not sufficient to say that “the applied approach allowed for the detection of the beneficial VOCs and taxa ..”. Instead, it is better to indicate what are the VOCs involved in this proposed synergistic effect (the real findings of the research), and what is the degree of support of such findings.
INTRODUCTION
Introductions refers to very diverse topics ranging from inflammasome, insulin resistance, glycogen synthase, biliary excretion defects, dysbiosis ... in an excessive number of 12 paragraphs. Paragraphs have little connection among them. Surprisingly, "diet and physical activity" (important variables according to the title of the manuscript) are only mentioned at the end of the introduction. Instead of showing all topics related to NAFLD (exhaustive list would be very long), it would have been better to integrate sentences in 4-5 paragraphs, capturing the essence of the contribution made by the research conducted by the authors. In opinion of this reviewer, I would have been better to explain why authors think that fecal microbiome may contribute to explain the effects of diet and physical activity on NAFLD.
In the introduction, a small description of the results of previously published trials is necessary to put the reader in context.
METHODS
STUDY GROUP
There is no information on the place of recruitment and period of time of recruitment …. The reader needs to know all these details to understand the paper.
It is not clear if the fecal samples come for the study in reference 18 or the study of references 19-20. Please, explain and clarify.
Are the fecal samples only obtained after the trial? If there are no samples obtained in the baseline of the trial, please discuss such limitation and implications in the Discussion section.
The sentence "All the co-authors had access to the study data and had reviewed and approved the final manuscript." is more suitable for a cover letter, rather than in the text of the manuscript (unless required by the journal).
Clearly, sections of 2.2 to 2.5 are the most solid part of the paper.
In methods, it is quite surprising that the words “Diet and physical activity” are not mentioned at all. No information on how diet or physical activity were monitored during the trial. Again, this is not concordant with the importance of these terms (“Diet and physical activity”) in the title of the manuscript.
DISCUSSION
Discussion mainly describes the factors found in the statistical analysis of metabolome, and later the correlation between taxa and VOCs. Diet and Physical activity again are only mentioned marginally.
To stablish a hierarchy of relevance in the findings will be a great improvement.
Include a paragraph discussing the limitations of the study.
CONCLUSION
As it occurs in the abstract, the conclusion is based on the usefulness of the approach. However, research (in general; except for methodological papers) is not designed to see whether an approach is useful or not, but to make findings (hopefully relevant findings). The challenge in this manuscript is to translate such big amount of information in true findings. Instead of focusing on the usefulness of the approach, it would be better to mention at least one or two findings that may/might be relevant in this research.
SUMMARY
This manuscript provides a lot of interesting metabolomic and microbiome information. However, it is necessary to put the research in the context of the title of the manuscript: “Synergistic effect of diet and physical activity on a NAFLD cohort”. If authors wish to maintain this focus, it is imperative to describe first the results from the intervention study carried out with diet and physical activity on NAFLD. Then, the reader will better understand the relevance of the subsequent metabolomic and microbiome analysis. It is necessary to try to identify the main finding(s) of the study and put them in the conclusion.
English language needs extensive editing
Author Response
Comments and Suggestions for Authors
ABSTRACT
It is surprising that the two main words of the title of the manuscript (“diet” and “physical activity”) are not in the aim of the abstract. As a reviewer, it is expected a correspondence between the title of the manuscript, the aim and the conclusion.
We thank the reviewer for her/his precious review and the raised issues. We are sorry for have omitted the connection with the title and the main topic of the paper. We extensively modified the abstract body.
In the abstract, there is no mention to the number of subjects involved in the analysis or their characteristics except saying “an Italian NAFLD patient cohort composed of patients following a physical activity intervention alone or coupled with the Mediterranean diet”.
Many thanks. We added the number of patients used in the study and we created a new sentence at lines 18-19
The acronym VOCs appears in the abstract without definition. VOCs it is not defined until the line 115 (Volatile Organic compounds). The term VOCs was used six times in the text before providing a definition of the acronym.
Thank you. We are sorry for have not declared the acronym the first time it is mentioned in the abstract. We modified the text accordingly (please see line 20).
The conclusion of the abstract is very vague, with no connection to the title of the manuscript. If the title of the manuscript is on the synergistic effect of diet and physical activity, it is expected to see in the conclusion whether this putative interaction is synergistic or not, and whether there are VOCs involved in such putative synergistic effects. According to this referee, it is not sufficient to say that “the applied approach allowed for the detection of the beneficial VOCs and taxa ..”. Instead, it is better to indicate what are the VOCs involved in this proposed synergistic effect (the real findings of the research), and what is the degree of support of such findings.
We thank the reviewer for highlighting this point. We in detail specified the abstract body specific finding on VOCs and taxa as follows: “In comparison with a physical activity regimen alone, in this study, we disclose how ethyl val-erate and pentanoic acid butyl ester, methyl valerate, and 5-hepten-2-one, 6-methyl changed be-cause of the positive synergistic effect exerted by the combination of the Mediterranean diet and physical activity regimens. Moreover, 5-hepten-2-one, 6-methyl positively correlated with Sanguinobac-teroides, as well as the two genera Oscillospiraceae-UCG002 and Ruminococcaceae UCG010 genera”.
INTRODUCTION
Introductions refers to very diverse topics ranging from inflammasome, insulin resistance, glycogen synthase, biliary excretion defects, dysbiosis ... in an excessive number of 12 paragraphs. Paragraphs have little connection among them. Surprisingly, "diet and physical activity" (important variables according to the title of the manuscript) are only mentioned at the end of the introduction. Instead of showing all topics related to NAFLD (exhaustive list would be very long), it would have been better to integrate sentences in 4-5 paragraphs, capturing the essence of the contribution made by the research conducted by the authors. In opinion of this reviewer, I would have been better to explain why authors think that fecal microbiome may contribute to explain the effects of diet and physical activity on NAFLD.
In the introduction, a small description of the results of previously published trials is necessary to put the reader in context.
We extensively revised the introduction section following the precious modification required by the reviewer. Specifically, i) we better explain the contribute of faecal microbiome downstream to diet and physical activity interventions (lines 67 to 70), ii) we reduced the number of short paragraphs and collapse the information in more synthetical sentences. Diet and physically activity concepts have been introduced and linked also in the other sections of the paper.
METHODS
STUDY GROUP
There is no information on the place of recruitment and period of time of recruitment …. The reader needs to know all these details to understand the paper.
Many thanks. We added the requested information at lines 86-90.
It is not clear if the fecal samples come for the study in reference 18 or the study of references 19-20. Please, explain and clarify.
Thank you. We better clarify how the three references are linked to the original NUTRIATT project and the subsequent dataset analyses made.
Are the fecal samples only obtained after the trial? If there are no samples obtained in the baseline of the trial, please discuss such limitation and implications in the Discussion section.
We thank the referee for pointing out this topic. We better specified that stool samples were retrieved both at 45 and 90 days of treatment (please see lines 111-113). Moreover we explained that being us interested in disclosing VOC between patients under a Mediterranean dietary regimen alone or combined with physical activity (both aerobic or anaerobic), the time variable was not considered for statistical stratification of samples (please see lines 113-117).
The sentence "All the co-authors had access to the study data and had reviewed and approved the final manuscript." is more suitable for a cover letter, rather than in the text of the manuscript (unless required by the journal).
Many thanks, we removed the sentence from the text.
Clearly, sections of 2.2 to 2.5 are the most solid part of the paper.
Thanks. We appreciated this reviewer comment.
In methods, it is quite surprising that the words “Diet and physical activity” are not mentioned at all. No information on how diet or physical activity were monitored during the trial. Again, this is not concordant with the importance of these terms (“Diet and physical activity”) in the title of the manuscript.
We thank the reviewer for her/his comment. In line with the required modifications, a new paragraph entitled “dietary intervention and data collection” has been added to the Material and Method section (please see lines 120-132).
DISCUSSION
Discussion mainly describes the factors found in the statistical analysis of metabolome, and later the correlation between taxa and VOCs. Diet and Physical activity again are only mentioned marginally.
To stablish a hierarchy of relevance in the findings will be a great improvement.
Thanks for this comment. We simplify the structure of the discussion section thus highlighting the finding with more relevance. At the end, we added a part resuming the most interesting findings found.
Include a paragraph discussing the limitations of the study.
Thanks. We reported the limitation of the study in the conclusion paragraph (please see lines 453-459).
CONCLUSION
As it occurs in the abstract, the conclusion is based on the usefulness of the approach. However, research (in general; except for methodological papers) is not designed to see whether an approach is useful or not, but to make findings (hopefully relevant findings). The challenge in this manuscript is to translate such big amount of information in true findings. Instead of focusing on the usefulness of the approach, it would be better to mention at least one or two findings that may/might be relevant in this research.
Many thanks. The conclusion section body was re-written thanks to the reviewer suggestions. Please see lines 441-464.
SUMMARY
This manuscript provides a lot of interesting metabolomic and microbiome information. However, it is necessary to put the research in the context of the title of the manuscript: “Synergistic effect of diet and physical activity on a NAFLD cohort”. If authors wish to maintain this focus, it is imperative to describe first the results from the intervention study carried out with diet and physical activity on NAFLD. Then, the reader will better understand the relevance of the subsequent metabolomic and microbiome analysis. It is necessary to try to identify the main finding(s) of the study and put them in the conclusion.
Comments on the Quality of English Language
English language needs extensive editing
Thank you. We extensively revised the text by improving the whole quality of our manuscript and, as you recommended, we submitted the paper to the language editing service (MDPI rapid English correction service - Certificate annexed).

Reviewer 2 Report
The article "Synergistic effect of diet and physical activity on a NAFLD cohort: metabolomics profile and clinical variable evaluation" provides knowledge for a very hot topic: lifestyle intervention and liver diseases. Nevertheless, there are some points that should be fixed:
The Introduction is mainly composed by the compilation of short paragraphs that are not linked with each other. Please, try to link them in a more fluently way.
In the sentence "Impaired liver uptake, conjugation and/or biliary excretion defects, are associated with altered bilirubin levels and liver lesions determining a decrease in the hepatocyte cell count, a condition that may also lead to hyperbilirubinaemia [14]", could you please clarify what does "hepatocyte cell count" mean?
The authors state that "Significantly, higher total bilirubin concentrations have been associated with higher liver stiffness values in alcohol-related liver disease patients, as well as have been found altered in NAFLD patients [15]." Nevertheless, citation 15 does not seem aproppiate in this case.
When describing the patient cohort, could the authors provide any severity score for NAFLD?
Analysis regarding the role of sex and age would improve the robustness of the article.
English language is good and easy to understand.
Author Response
Comments and Suggestions for Authors
The article "Synergistic effect of diet and physical activity on a NAFLD cohort: metabolomics profile and clinical variable evaluation" provides knowledge for a very hot topic: lifestyle intervention and liver diseases. Nevertheless, there are some points that should be fixed:
The Introduction is mainly composed by the compilation of short paragraphs that are not linked with each other. Please, try to link them in a more fluently way.
We thank the reviewer for have raising this issue. To make the introduction points well connected, we extensively revised the whole section. Specifically, i) we better explain the contribute of faecal microbiome downstream to diet and physical activity interventions, ii) we reduced the number of short paragraphs and collapse the information in more synthetical sentences.
In the sentence "Impaired liver uptake, conjugation and/or biliary excretion defects, are associated with altered bilirubin levels and liver lesions determining a decrease in the hepatocyte cell count, a condition that may also lead to hyperbilirubinaemia [14]", could you please clarify what does "hepatocyte cell count" mean?
Many thanks. We are sorry for have not detailed the meaning before. We meant that the cell counts were obtained after biopsy and microscope inspection. Anyway, after collapsing different sentences as required by the other reviewer, we modified this part in the text.
The authors state that "Significantly, higher total bilirubin concentrations have been associated with higher liver stiffness values in alcohol-related liver disease patients, as well as have been found altered in NAFLD patients [15]." Nevertheless, citation 15 does not seem aproppiate in this case.
We are really sorry for the wrong reference. We change the cited paper in the bibliography and the sentence was also modified according to other raised issues.
When describing the patient cohort, could the authors provide any severity score for NAFLD?
Many thanks. We better describe in the material and method section how we measured the CAP score. Please see lines 90-94
Analysis regarding the role of sex and age would improve the robustness of the article.
We thank the reviewer for her/his useful and precise comment. The stratification of samples based on other metadata including age and sex has been inspected by using the discriminant analysis of principal component and other approaches including two group comparison statistics. Not a clear differential distribution of groups emerged thus indicating how these variables have low impact on our patient cohort. We aim at enlarging sample number and conduct a multi-centric study and for sure all the variables will be tested again for their impact.

Round 2
Reviewer 1 Report
Authors responded adequately to my comments.
Reviewer 2 Report
All my comments have been addressed. Thank you very much for the effort.
English is understandable.